# Risk and Protective Factors and Interventions for Reducing Juvenile Delinquency: A Systematic Review

Aida Aazami [1], Rebecca Valek [2], Andrea N. Ponce [2] and Hossein Zare [2,3,*]

1   School of Behavioral and Brain Sciences, The University of Texas at Dallas, Dallas, TX 75080, USA; aida.aazami@gmail.com
2   Department of Health Policy and Management, Johns Hopkins Bloomberg School of Public Health, Baltimore, MD 21205, USA; rvalek1@jhmi.edu (R.V.); aponce4@jhu.edu (A.N.P.)
3   Global Health Services and Administration, The School of Business, University of Maryland Global Campus, Adelphi, MD 20774, USA
*   Correspondence: hzare1@jhu.edu

**Abstract:** Juvenile delinquency is a pressing problem in the United States; the literature emphasizes the importance of early interventions and the role of the family in preventing juvenile delinquency. Using the Preferred Reporting Items for Systematic Reviews and Meta-Analyses (PRISMA) framework, PudMed, and Scopus, we included 28 peer-reviewed articles in English between January 2012 and October 2022. We evaluated the existing literature regarding the risk factors, protective factors, and interventions related to juvenile delinquency. We searched articles that discussed reducing juvenile delinquency and recidivism in the U.S. and coded them into four overarching themes: 'family conflict and dysfunction', 'neglect and maltreatment', 'individual and family mitigating factors', and 'family- and community-based interventions'. We found that family conflict and dysfunction and neglect and maltreatment were two primary predictors of juvenile delinquency. Notably, higher academic achievement and strong and positive parental relationships were factors that protected against delinquency amongst at-risk youth. Interventions that yielded optimal efficacy in curbing recidivism included family-based interventions, specifically family therapy, and community-based interventions. Considering multi-dimensional factors that affect delinquent behaviors, interventions should consider the influence of family, peers, neighborhood, schools, and the larger community.

**Keywords:** juvenile delinquency; prevention; family dynamics; intervention

## 1. Introduction

Juvenile delinquency is a major ongoing issue that plagues many communities in the United States (U.S.), with an estimated 424,300 arrests involving persons younger than 18 years old occurring in 2020 (Puzzanchera 2022). The most common crimes for which juveniles are incarcerated include theft, assault, vandalism, drug abuse violations, and violence (OJJDP 2020). Although substantial progress has been made, the U.S. has a higher rate of youth incarceration than any other developed nation (Barnert et al. 2017). Racial and ethnic disparities in the juvenile justice system further distinguish the U.S. from other countries; youth of color make up 68% of detained youth, yet they make up 44% of the U.S. adolescent population (Amani et al. 2018).

Youth who are incarcerated face higher rates of violence, fewer prospects for education and employment, poorer health outcomes, and a higher likelihood of returning to prison as an adult (Amani et al. 2018). About 80% of the youth who are arrested annually face reincarceration as adults (Barnert et al. 2017), and experiences with long-term incarceration frequently result in recidivism and persistent criminal behavior (Amani et al. 2018). These discouraging outcomes from juvenile incarceration come at the high average cost of approximately $214,000 per person per year plus potential costs of future recidivism and lost future earnings (Sanders 2021). Public opinion polling shows strong and widespread

support for alternatives to incarceration for juvenile offenders, with a desired emphasis on rehabilitation (The PEW Charitable Trusts 2014). It is more economically and socially feasible to focus on prevention and rehabilitation.

Family- and community-based alternatives have shown success in reducing recidivism for youth, even those who commit serious and violent crimes (Underwood and Washington 2016). Several intervention models have been developed to address the issue of juvenile delinquency. Multi-systemic therapy takes a systemic approach, targeting various systems surrounding youth (family, school, and community) to address risk factors and enhance protective factors for delinquency prevention (Vidal et al. 2017). Multi-systemic therapy may be enhanced with the integration of contingency management, a form of behavioral therapy based on reinforcement and rewards (Petry et al. 2017). These intervention models offer unique strategies for addressing the complex needs of at-risk youth and their environments. Functional Family Therapy is a family-based intervention that targets dysfunctional family dynamics and aims to improve communication, problem-solving, and conflict resolution skills (D.C. Department of Human Services n.d.; Gan et al. 2019). Multidimensional Family Therapy focuses on individual and family factors, combining cognitive–behavioral and ecological system approaches to address delinquency risk factors comprehensively (Hogue et al. 2006; U.S. Department of Justice Office of Justice Programs 2012). Adolescent group treatment involves peer-oriented group therapy, emphasizing social skills development, emotional regulation, and positive peer influences (Arias-Pujol and Anguera 2017). Juvenile courts have begun to shift toward increased use of these alternatives, with 43% of cases being redirected to community-based services as of 2018 (Buchanan et al. 2020). This shift requires an improved understanding of existing interventions to reduce juvenile delinquency and their effectiveness, including understanding how families can be empowered to serve as protective factors against juvenile delinquency (Amani et al. 2018). In addition, understanding the risk and protective factors that may contribute to juvenile delinquency enables the creation of more tailored interventions that target these factors.

This systematic review aims to further understand the landscape of juvenile delinquency by providing a synthesized analysis of the factors that both contribute to and mitigate against it. Additionally, the review evaluates the efficacy of existing family- and community-based interventions and the role of families and communities in fostering positive outcomes for at-risk youth. Furthermore, this review contributes to the current literature on interventions that can extenuate the factors contributing to recidivism amongst youth. We will complement articles identified in this review with a bias analysis to enhance credibility and transparency of our findings.

## 2. Materials and Methods

This systematic review utilized the Preferred Reporting Items for Systematic Reviews and Meta-Analyses (PRISMA) framework and the Synthesis Without Meta-analysis (SWiM) reporting guidelines (Campbell et al. 2020; Moher et al. 2009; Page et al. 2021). The PRISMA framework outlines four steps for determining articles to be used in a systematic review: identification, screening, eligibility, and inclusion. This review was conducted from October 2022 through January 2023.

### 2.1. Inclusion Criteria

Peer-reviewed articles published in English between January 2012 and October 2022 that discussed reducing juvenile delinquency and recidivism in the U.S. and related protective and risk factors were included in the review. In particular, we included articles with a primary focus on familial dynamics, mitigating factors, and family-based interventions, including those that extended into the broader community, and their impacts on juvenile delinquency and recidivism. Observational, experimental, qualitative, and quantitative studies that met these criteria and did not meet any exclusion criteria were included in the review. We detailed inclusion and exclusion criteria based on the PICO guideline (Population, Intervention, Comparison, and Outcomes) in Table 1.

**Table 1.** The systematic review inclusion and exclusion criteria based on the PICO guideline.

| Criteria | Notes |
| --- | --- |
| **Inclusion criteria** | |
| Participants | - Any studies that sampled families, parents, guardians, or siblings or examined factors at the household level (familial dynamics). <br> - Any studies that examined factors or attributes that reduce the risk of recidivism or delinquency or factors that could be targeted for interventions (mitigating factors). <br> - Any studies that examined household-level strategies, programs, or interventions aimed at preventing or reducing recidivism and delinquency, including those that extend into the broader community, and their impacts on juvenile delinquency and recidivism (family-based interventions). |
| Intervention | The focus of the study was family-based interventions. <br> - Any studies that examined household-level strategies, programs, or interventions aimed at preventing or reducing recidivism and delinquency |
| Comparators | Any studies with any comparator included. |
| Outcomes | We included any studies of interventions meeting the above criteria to determine the proportion that reported engagement outcomes |
| Study design | Observational, experimental, qualitative, and quantitative studies that met these criteria and did not meet any exclusion criteria were included in the review. |
| Exclusion criteria | |
| Participants | - Studies included conduct disorder, internalizing and externalizing symptoms, and substance abuse. <br> - Studies that focused on the siblings or parents of juvenile offenders and on justice system, welfare system, or court policies—as opposed to the use of family interventions within these systems or risk and mitigating factors of individuals involved with these systems—were determined to be outside of the scope of this review. |
| Intervention | Interventions with a primary focus other than family-based interventions. |
| Study design | Systematic reviews, literature reviews, and meta-analyses |

### *2.2. Exclusion Criteria*

Articles were excluded if research was: (1) conducted outside of the U.S., (2) focused on an outcome irrelevant to the study or outside of the scope of reducing juvenile delinquency, (3) nonempirical, or (4) not published in English. Topics that warranted exclusion included conduct disorder, internalizing and externalizing symptoms, and substance abuse. Studies that focused on the siblings or parents of juvenile offenders and on the justice system, welfare system, or court policies—as opposed to the use of family interventions within these systems or risk and mitigating factors of individuals involved with these systems—were determined to be outside of the scope of this review. Finally, systematic reviews, literature reviews, and meta-analyses were excluded from the final analysis.

### *2.3. Data Sources and Search Strategy*

We conducted a literature search using two electronic databases: Scopus and PubMed. The Scopus database covered 72% of published articles; the PubMed database was used in conjunction to fill in the gaps left by Scopus (Aksnes and Sivertsen 2019). Table 2 displays the search strategies used in each database, respectively. Figure 1 outlines our process of selecting articles for review following the PRISMA framework (Moher et al. 2009).

**Table 2.** Search strategy.

| Electronic Database | Search Strategy |
| --- | --- |
| Scopus | ("juvenile delinquency" OR "juvenile crime") AND (("family intervention")) AND (psychological) OR (mental AND health) OR (psychology) OR (police) AND (LIMIT-TO (LANGUAGE, "English")) |
| PubMed | (((Juvenile delinquency) AND (family intervention OR family OR "family-based")) AND (psychological OR mental OR psychology OR "mental health")) AND (crime OR police) |

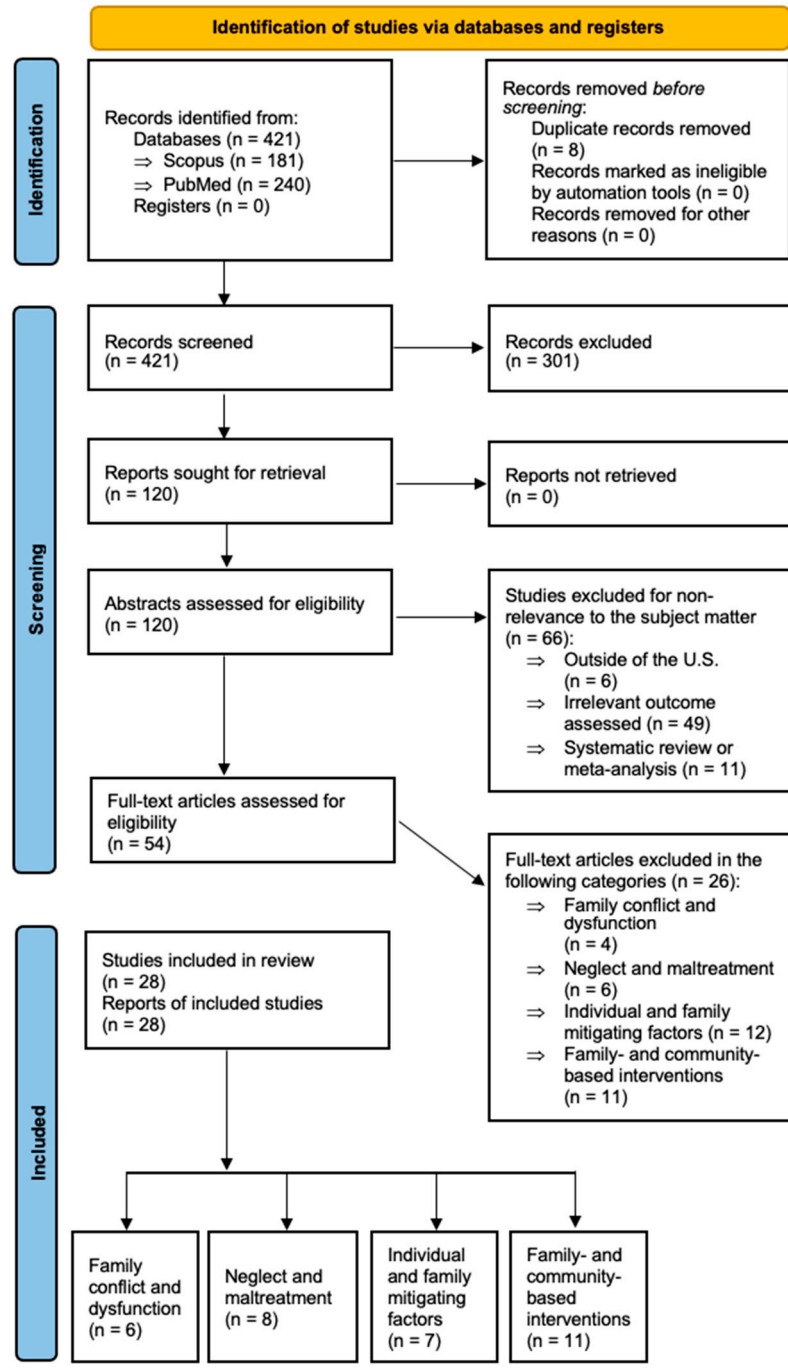

**Figure 1.** PRISMA flow diagram identifying articles for inclusion in qualitative analysis.

Our initial search yielded 421 unique citations (181 from Scopus and 240 from PubMed). Three authors (A.A., R.V., and A.N.P.) independently screened the titles, abstracts, and full texts using the pre-determined eligibility criteria. At each stage, an article was included if it was selected for inclusion by at least two reviewers. In the case of ambivalence, the senior author (H.Z.) was consulted to reach a final consensus.

Twenty-eight of these full-text articles were selected for inclusion in the qualitative synthesis based on relevance and exclusion criteria and were then reviewed for content analysis and synthesized. The citation manager EndNote (EndNote™ 20) was used to organize the selected references.

### 2.4. Risk of Bias Assessment

We followed the Cochrane Handbook for Systematic Reviews (Higgins et al. 2022). Included articles were evaluated for the risk of bias. We performed a domain-based evaluation for each study across five domains: selection bias, performance bias, detection bias, attrition bias, reporting bias, and other biases, including biases involving data quality and analysis methods. Judgment of studies for potential bias was indicated by assigning 'low risk', 'high risk, or 'unclear risk' for each respective source of bias. The Cochran RevMan was used to present reviewers' assessments and prepare aggregated risk of bias evaluation plots. The evaluation was conducted independently by two reviewers.

### 3. Results

The title screening process eliminated 301 articles based on relevance or exclusion criteria. We obtained the abstracts of the remaining 120 articles and subsequently eliminated an additional 66 articles through abstract screening. Full texts were obtained for the remaining 54 articles, which were divided among the three reviewers and assessed for eligibility for inclusion in the qualitative synthesis based on relevance and exclusion criteria. Based on the full text review, we developed the following themes: (1) family conflict and dysfunction, (2) neglect and maltreatment, (3) individual and family mitigating factors, and (4) family- and community-based interventions. Full text articles were categorized into these themes and synthesized. In Table 3, we have reported study design, data sources, study populations, and key results from each study.

**Table 3.** Findings of the study included in this systematic review.

| Study | Study Population | Outcome(s) Measured | Principal Findings |
|---|---|---|---|
| **Family Conflict and Dysfunction** | | | |
| Trinkner et al. (2012) | Middle and high school students in New Hampshire participating in the New Hampshire Youth Study from 2007–2009 (n = 596) | Delinquency and parental legitimacy | Authoritative parenting is positively and authoritarian parenting is negatively associated with parental legitimacy. Parental legitimacy reduces the likelihood of future delinquency. |
| Sitnick et al. (2017) | Low-income males living in an urban community followed from ages 18 months through adolescence (15–18 years) (n = 310) | Juvenile petitions from juvenile court records | Early-childhood individual and family factors (such as harsh parenting and poor emotional regulation) can discriminate between adolescent violent offenders and nonoffenders or nonviolent offenders. |
| Lippold et al. (2018) | Early adolescents in two-parent homes and their parents (n = 618) in Iowa and Pennsylvania. PROSPER study | Youth substance use and delinquency in 9th grade | Changes in the parent–youth relationship, such as decreased parental warmth and increased hostility during adolescence, were associated with increased delinquency, especially for girls. |

**Table 3.** *Cont.*

| Study | Study Population | Outcome(s) Measured | Principal Findings |
|---|---|---|---|
| Mowen and Boman (2018) | Male youth (under age 18) and "youthful offenders" (under age 25 and incarcerated under "Youthful Offender" laws) across Colorado, Florida, Kansas, and South Carolina (n = 337) Serious and Violent Offender Reentry Initiative youth sample collected 2005–2007 | Crime and substance use | Family conflict is a major driver of recidivism through its direct impact on increasing crime and substance use and more reentry programs focused on reducing family conflict should be explored, such as multisystemic therapy. |
| Anderson and Walerych (2019) | Qualitative study; Juvenile court officers working with girls in the juvenile justice system (n = 24) | Extent and type of trauma experienced by girls in the juvenile justice system | In qualitative interviews, the officers discussed how exposure to trauma (violence at home, a dysfunctional home, etc.) influenced girls' trajectory and contributed to many of their involvement with the juvenile justice system. |
| Garduno (2022) | Adolescents attending public middle or high school in Maryland receiving services from Identity, Inc. (n = 555) | Three deviant behaviors: stealing, fighting, and smoking marijuana | Experience of multiple adverse childhood experiences increased the likelihood of adolescents engaging in deviant behaviors. School connection, anger management skills, and parental supervision acted as protective factors. |
| **Neglect and Maltreatment** | | | |
| Ryan (2012) | Youth ages 8–16 who had their first episode in a substitute child care welfare setting between 2000–2003 in the state of Washington (n = 5528) | Risk of justice involvement | Youth with behavioral problems were more likely to be placed in congregate care facilities and had little access to family-based services. High arrest rates among youth with behavioral problems indicated an ineffectiveness of the congregate care approach. |
| Ryan et al. (2013) | Moderate and high-risk juvenile offenders who were screened for probation from 2004–2007 in Washington (n = 19,833) | Risk of subsequent offending (based on event history models) | Returning to an environment where one faced continued or ongoing neglect increased an individual's risk of re-offending. |
| Logan-Greene and Jones (2015) | Youth who were assessed at age 14 at one of the five study sites across the U.S. in the LONGSCAN consortium (n = 815) | Aggression and delinquency | Experiencing chronic neglect or chronic failure to provide from ages 0–12 was associated with increased aggression and delinquency at age 14. This relationship was mediated by social problems, especially for girls. |
| Ezell et al. (2018) | Court staff across four rural juvenile courts in Michigan (n = 15) | Qualitative interviews on trauma-informed practice | Court staff widely supported trauma-informed practices like mental health referrals instead of—or in addition to—sentencing or punishment but faced challenges due to limited mental health resources and inadequate support from schools, government, and police. |

**Table 3.** *Cont.*

| Study | Study Population | Outcome(s) Measured | Principal Findings |
|---|---|---|---|
| Lantos et al. (2019) | U.S. adolescents enrolled in grades 7–12 from 1994–95 (n = 10,613) National Longitudinal Study of Adolescent Health | Violent and nonviolent offending behavior | Experiences of maltreatment were associated with more rapid increases in both non-violent and violent offending behaviors. |
| Wilkinson et al. (2019) | U.S. adolescents enrolled in grades 7–12 from 1994–95 (n = 10,613) National Longitudinal Study of Adolescent Health | Violent and non-violent offending frequency | High-quality relationships with mother or father figures, school connection, and neighborhood collective efficacy were protective against violent offending (both for those experiencing and not experiencing maltreatment). |
| Logan-Greene et al. (2020) | Medium- to high-risk youth on probation (n = 5378) Washington State Juvenile Assessment | Self-regulation, mental health, substance use, academic functioning, family/social resources, and behavioral problems | Groups of individuals exposed to different adverse childhood experiences varied in terms of all six outcomes, suggesting a need for more differentiated treatment approaches applied early on to address these unique needs. |
| Garduno (2022) | Adolescents attending public middle or high school in Maryland receiving services from Identity, Inc. (n = 555) | Three deviant behaviors: stealing, fighting, and smoking marijuana | Experience of multiple adverse childhood experiences increased the likelihood of adolescents engaging in deviant behaviors. School connection, anger management skills, and parental supervision acted as protective factors. |
| **Individual and Family Mitigating Factors** | | | |
| Ryan (2012) | Youth ages 8–16 who had their first episode in a substitute child care welfare setting between 2000–2003 in the state of Washington (n = 5528) | Risk of justice involvement | Youth with behavioral problems were more likely to be placed in congregate care facilities and had little access to family-based services. High arrest rates among youth with behavioral problems indicated an ineffectiveness of the congregate care approach. |
| Halgunseth et al. (2013) | Rural adolescents and their parents (n = 342 adolescents) in Iowa and Pennsylvania. 6-year PROSPER (PROmoting School-community-university Partnership to Enhance Resilience) study. | Delinquent-oriented attitudes, deviant behaviors (stealing, carrying a hidden weapon, etc.) | Inconsistent discipline at home may lead adolescents to develop accepting attitudes toward delinquency, which may contribute to future antisocial and deviant behaviors. |
| Cavanagh and Cauffman (2017) | Low- to moderate-level male offenders ages 13–17 who participated in the Crossroads study of first-time juvenile offenders and their mothers conducted in California, Louisiana, and Pennsylvania (n = 634, or 317 mother–son pairs) | Re-offending | Strong mother–son relationships can serve as a protective factor against youth's re-offending, especially for older youth. |
| Robst et al. (2017) | Youth involved with the Florida juvenile justice system from July 2002–June 2008 with records of 'severe emotional disturbance' and an out-of-home placement following arrest (n = 1511) | Re-arrest during a 12-month period | Severe trauma history increased the likelihood of re-arrest relative to less severe or no trauma history. Among those with severe trauma history, those placed in foster homes had the lowest rates of recidivism compared to other out-of-home placements. |

**Table 3.** *Cont.*

| Study | Study Population | Outcome(s) Measured | Principal Findings |
|---|---|---|---|
| Ruch and Yoder (2018) | 10–20-year-old youth in custody in the U.S. (n = 7073) Survey of Youth in Residential Placement | Likelihood of having a plan for education and employment after reentry | Family contact during incarceration increased the likelihood that youth had educational and employment reentry plans. |
| Wilkinson et al. (2019) | U.S. adolescents enrolled in grades 7–12 from 1994–95 (n = 10,613) National Longitudinal Study of Adolescent Health | Violent and non-violent offending frequency | High quality mother or father relationships, school connections, and neighborhood collective efficacy were protective against violent offending (both for those experiencing and not experiencing maltreatment). |
| Gearhart and Tucker (2020) | Mothers with children of at least 13 years of age and born in 20 select U.S. cities (n = 3444 families) Fragile Families and Child Wellbeing Study | Self-reported juvenile delinquency | Individual-level factors are stronger predictors of self-reported juvenile delinquency than collective efficacy. Mitigating factors include satisfaction with school, academic performance, and parental closeness. Risk factors include substance use, delinquent peers, impulsivity, and prior delinquency. |
| **Family- and Community-Based Interventions** | | | |
| Henggeler et al. (2012) | Juvenile offenders ages 12–17 engaged in one of six juvenile drug courts participating in the study (n = 104) | Marijuana use and crime | The use of contingency management in combination with family engagement strategies was more effective than the usual treatment at reducing marijuana use, crimes against persons, and crimes against property among juvenile offenders. |
| Trinkner et al. (2012) | Middle and high school students in New Hampshire participating in the New Hampshire Youth Study from 2007–2009 (n = 596) | Delinquency and parental legitimacy | Authoritative parenting is positively associated with and authoritarian parenting is negatively associated with parental legitimacy. Parental legitimacy reduces the likelihood of future delinquency. |
| White et al. (2013) | Previously arrested youth ages 11–17 who participated in a functional family therapy program (n = 134) | Post-treatment levels of adjustment and likelihood of offending | Individuals with callous-unemotional traits face more challenges and symptoms when beginning treatment and are more likely to violently offend during treatment, but functional family therapy can help to reduce their likelihood of violent offending post-treatment. |
| Bright et al. (2014) | Youth ages 11–19 with a history of juvenile justice involvement receiving intensive in-home services from 2000–2009 in the Southeastern United States (n = 5000) | Classification of youth as recidivists, at-risk, or non-recidivists | The model of in-home services was associated with reduced re-offending, particularly among girls, and with increased likelihood of living at home and attending or completing school for both boys and girls. |
| Dakof et al. (2015) | Youth ages 13–18 participating in a juvenile drug court in Florida (n = 112) | Offending and substance use | The results support the use of family therapy in juvenile drug court treatment programs to reduce criminal offending and recidivism. |

**Table 3.** *Cont.*

| Study | Study Population | Outcome(s) Measured | Principal Findings |
|---|---|---|---|
| Barrett and Janopaul-Naylor (2016) | Active cases of youth ages 10–17 involved with the Safety Net Collaborative in Cambridge, Massachusetts, in 2013 (n = 30) | Arrest rates and mental health referrals | Following the implementation of the safety net collaborative, an integrated model that provides mental health services for at-risk youth, community arrest rates declined by over 50%. |
| Karam et al. (2017) | Moderate- to high-risk juvenile offenders involved in the Parenting with Love and Limits group and family therapy program between April 2009 to December 2011 in Champaign County, Illinois (n = 155 in treatment; n = 155 in control group) | Recidivism rates and parent-reported behavior | The Parenting with Love and Limits group and family therapy program was associated with significantly reduced recidivism rates and behavioral improvements, indicating potential effectiveness of family and group therapy to reduce recidivism among those at the highest risk. |
| Vidal et al. (2017) | Rhode Island youth participating in a multisystemic therapy program (n = 577) and in a control group (n = 163) | Out-of-home placement, adjudication, placement in a juvenile training school, and offending | Receipt of multisystemic therapy was associated with lower rates of offending, out-of-home placement, adjudication, and placement in a juvenile training school, demonstrating the potential efficacy of multisystemic therapy in reducing delinquency among high-risk youth. |
| D'Agostino et al. (2020) | ZIP codes with the Fit2Lead park-based violence prevention program and matched control communities without the program in Miami-Dade County, Florida from 2013–2018 (n = 36 ZIP codes) | Change in arrest rates per year among youth ages 12–17 | Park-based violence prevention programs such as Fit2Lead may be more effective at reducing youth arrest rates than other after-school programs. Results support the use of community-based settings for violence interventions. |
| Anderson et al. (2021) | Court-involved girls on probation from 2004–2014 in one Midwest juvenile family court who received the family-based intervention (n = 181) or did not (n = 803) | Recidivism rates | One-year recidivism rates were lower among girls who participated in the family-based intervention program compared to those just on parole. Qualitative interviews highlighted the importance of family-focused interventions for justice-involved girls. |
| Borduin et al. (2021) | Individuals involved in the Missouri Delinquency Project from 1990–1993 and randomized to multisystemic therapy for potential sexual behaviors or the usual treatment of cognitive behavioral therapy (n = 48) | Arrest, incarceration, and civil suit rates in middle adulthood | Participants assigned to the multisystemic therapy treatment were less likely to have been re-arrested by middle adulthood and had lower rates of sexual and nonsexual offenses, demonstrating the potential benefits of targeted therapies. |

Figures 2 and 3 display the risk of bias plots to show the distributions of each type of bias identified across all studies. Our assessment showed that all selected articles had a high risk of at least one type of bias, including "Other" biases (e.g., residual confounding, unaccounted cohort effects, misclassification of exposure, and bias due to missing data). Approximately 21% of the selected articles were at high risk of biases, with selection, reporting, and performance biases being the most frequent types of biases identified. See Figures 2 and 3 for more details.

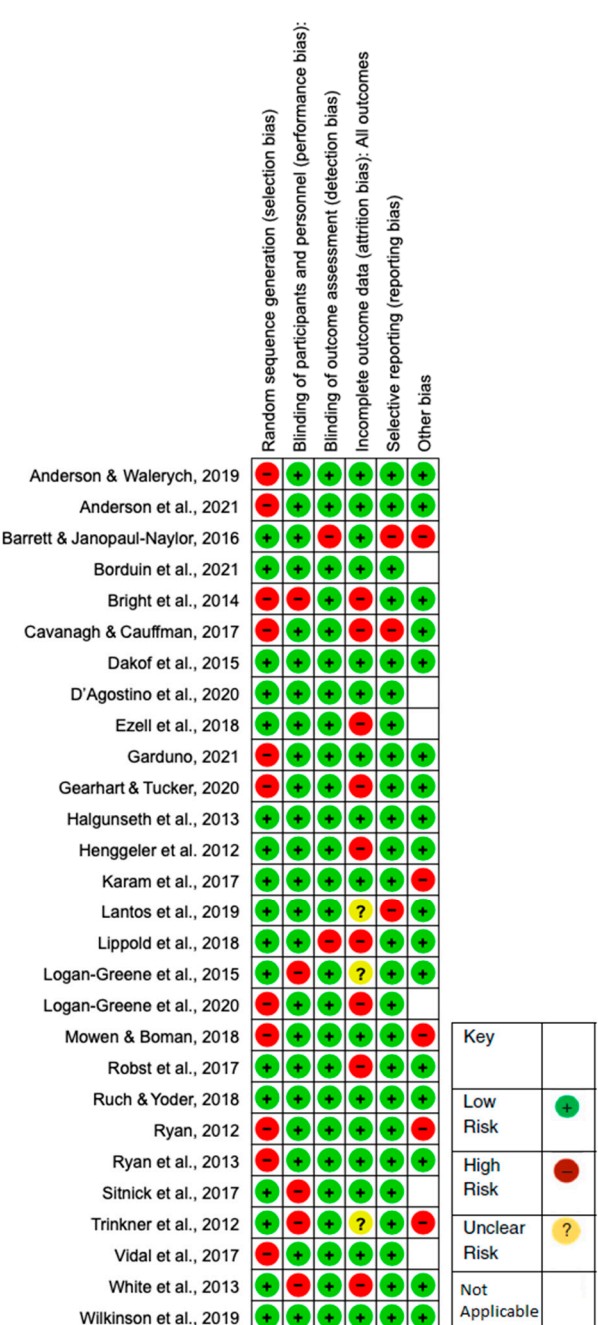

**Figure 2.** Risk of bias table.

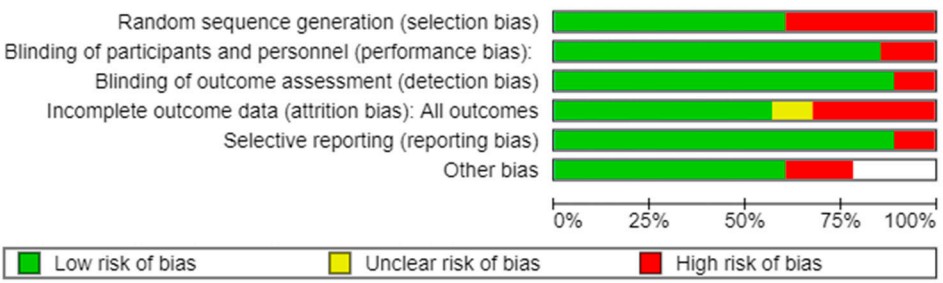

**Figure 3.** Risk of bias graph (Note: Blank space stands for Not Applicable).

**Family conflict and dysfunction.** All six selected articles that related to this theme recognized the important role of family in youth development. More specifically, problematic



family environments, such as family-based violence, adverse childhood experiences, and harsh parenting, have been reported as common exposures among youth that exhibited deviant behaviors (Anderson and Walerych 2019; Garduno 2022; Sitnick et al. 2017). The effect of parent–youth conflicts has been shown to yield different outcomes in boys and girls, and interventionists should be aware of how these consequences may vary depending on sex (Lippold et al. 2018). Girls appear to be more vulnerable to fluctuations in their parents' behaviors (i.e., switching between warmth and hostility) than boys, potentially due to the increased relationship orientation of girls (Lippold et al. 2018). The impact of family conflict continues to be relevant following incarceration, as conflict upon release has been shown to be significantly associated with higher levels of criminal reoffending (Mowen and Boman 2018). Overall, the articles highlight family conflict and dysfunction as a significant risk factor for juvenile delinquency, before and after experiencing incarceration, and suggest that supportive, positive parent–child relationships may protect against delinquency (Anderson and Walerych 2019; Mowen and Boman 2018; Sitnick et al. 2017; Trinkner et al. 2012).

**Neglect and maltreatment.** Neglect and maltreatment during childhood were found to be risk factors for delinquent behavior across eight articles. While one study showed a direct effect of neglect and maltreatment on increased offending (Lantos et al. 2019), others examined their contributions to adverse childhood experiences, for which a persistent association with increased engagement in delinquent or criminal behavior was found (Garduno 2022; Logan-Greene et al. 2020). Young victims of neglect—the most common form of chronic maltreatment—were significantly more likely to continue offending compared to youth who did not have a history of neglect, especially if returning to the environment in which they experienced neglect (Logan-Greene and Jones 2015; Ryan et al. 2013). Neglect and maltreatment may lead to placement in substitute care child welfare settings, which has been identified as a significant predictor of arrest, especially among youth with behavioral problems (Ryan 2012). The pathway between childhood maltreatment and adolescent offending can be disrupted through high quality parental relationships in addition to the adoption of trauma-informed practices in the justice system (Ezell et al. 2018; Wilkinson et al. 2019).

**Individual and Family Mitigating Factors.** We identified seven articles in this category. Our findings showed that school satisfaction, higher academic achievement, and parental closeness have been associated with reductions in delinquent and offending behaviors (Cavanagh and Cauffman 2017; Gearhart and Tucker 2020; Robst et al. 2017). However, the protective effect of connection to school was found to weaken over time for youth with experiences of maltreatment as compared to those without such experiences (Wilkinson et al. 2019). In this same study, a high-quality relationship, whether with a mother or father figure, was associated with a decrease in violent offending behaviors (Wilkinson et al. 2019). In contrast, youth who experienced inconsistent parental discipline and placement in a substitute care child welfare setting exhibited worse behavioral and criminal justice outcomes, respectively (Halgunseth et al. 2013; Ryan 2012). When experiencing incarceration, family remains influential, as continued family contact during incarceration increased the likelihood that young adults would continue their education or find a job upon release (Ruch and Yoder 2018). One study showed that although parents, caretakers, teachers and neighbors may have different roles depending on youth's home environments and experiences of trauma, their engagement may generally help reduce recidivism among youth (Robst et al. 2017).

**Family- and community-based interventions.** Based on 11 identified articles in this category, family- and community-based interventions showed promising results as evidence-based interventions for addressing delinquency, with an emphasis on interventions involving therapy. Multi-systemic therapy, an intensive treatment process involving youth, families, therapists, and case workers to address problem behaviors, has been found to reduce the risk of out-of-home placement, adjudication, and training school placement (Dakof et al. 2015; Vidal et al. 2017). Another study compared two types of family therapy:

multidimensional family therapy, which extends the treatment domain to include youth, family, and community, and adolescent group treatment, which involves therapist-led cognitive–behavioral therapy and motivational interviewing in a group setting (Dakof et al. 2015). Youth in multidimensional family therapy had lower frequencies of serious delinquent behaviors and felony arrests compared to youth in adolescent group treatment (Dakof et al. 2015). One study found that youth involved in juvenile drug courts with contingency management–family engagement intervention, which combined components of multi-systemic therapy and contingency management, had a 53% decrease in the rate of general delinquency as compared to a 14% increase in the rate of general delinquency among youth in juvenile drug courts receiving usual services (Henggeler et al. 2012). Interventions may improve outcomes for certain groups more than others; for example, one study found evidence to support that girls had lower rates of recidivism compared to boys after receiving intensive in-home services (Bright et al. 2014). Outcomes may additionally be dependent on individual traits and characteristics and types of offenders. Despite improved outcomes through functional family therapy, youth with callous–unemotional traits, such as lack of empathy, were shown to be at higher risk for violent offending during this treatment than other groups (White et al. 2013). Adaptions of multisystemic therapies may be tailored to youth who have committed sexual offenses and exhibit problematic sexual behaviors (Borduin et al. 2021). The effectiveness of these therapies may be improved by addressing parenting style and legitimacy and utilizing a multiple family group approach (Karam et al. 2017; Trinkner et al. 2012). Findings of these studies align with interviews with practitioners who point to family as an important consideration in interventions addressing juvenile delinquency and incarceration given their proximal influence on youth (Anderson et al. 2021).

While our search criteria focused on family-based interventions, family-based interventions commonly consider community contexts and may be implemented in larger community settings. Two articles included in our review highlighted specific community-based interventions. A spatial analysis conducted in Miami-Dade County, Florida, found that neighborhoods with a park-based afterschool program that focused on preventing violence and promoting mental health reduced youth arrest rates (D'Agostino et al. 2020). The Safety Net Collaborative in Cambridge, Massachusetts, a community-based intervention that integrates mental health professionals, police departments, schools, and human services departments to provide mental health treatment to at-risk youth, observed a more than 50% decrease in community arrests, providing support for the efficacy of multi-sectoral, community-based interventions (Barrett and Janopaul-Naylor 2016). Overall, these studies supported the benefits of both family- and community-based interventions in reducing juvenile delinquency and call for increased evaluation and wider implementation of the most effective interventions (Barrett and Janopaul-Naylor 2016; Dakof et al. 2015).

## 4. Discussion

In this systematic review, we reviewed 28 articles from the literature on risk and protective factors and interventions around juvenile delinquency using two data sources: Scopus and PubMed. Using the PRISMA framework and the SWiM reporting guidelines, 28 articles were synthesized across four predominant themes: family conflict and dysfunction, neglect and maltreatment, individual and family mitigating factors, and family- and community-based interventions. The articles reviewed in the three risk and protective factors categories—family conflict and dysfunction, neglect and maltreatment, and individual and family mitigating factors—highlighted factors that are protective against juvenile delinquency such as strong familial relationships and risk factors like traumatic experiences and familial dysfunction (Halgunseth et al. 2013; Lantos et al. 2019; Logan-Greene et al. 2020; Ryan 2012). The articles on family- and community-based interventions emphasized the importance of engaging the family, the potential efficacy of family-based therapies, and the need to evaluate and expand such interventions further (Dakof et al. 2015; Karam et al.

2017). The following sections explore the four main themes highlighted in this study in more detail.

**Family Conflict and Dysfunction.** Studies documented that family conflict and dysfunction can have a significant impact on juvenile delinquency (Anderson and Walerych 2019; Garduno 2022; Sitnick et al. 2017). Travis Hirschi's social control theory posits that juvenile delinquency is associated with the failure to develop strong social bonds with parents and friends and with the lack of participation in academic and social communities (Costello and Laub 2020; Wiatrowski and Swatko 1979). According to this theory, social bonds, such as families, play an essential role in reducing an individual's tendency to commit crime through the development of social norms of acceptable behavior (Costello and Laub 2020; Wiatrowski and Swatko 1979). A youth's family serves as a critical social institution that aids in the development of their behavior through modeling and reinforcing certain behaviors deemed socially acceptable (Costello and Laub 2020). Social learning theory postulates that interactions with others play a role in the development, maintenance, or modification of both criminal and conforming behaviors (Triplett 2007). Juveniles who are more grounded in familial ties are more likely to behave in a manner that aims to appease their parents (Hoeve et al. 2012). Strong parental attachment in young people increases their likelihood of caring about their parents' norms, which deters criminal urges (Hoeve et al. 2012). Conversely, weak parent–child relationships are associated with increases in delinquent behaviors in both boys and girls (Hoeve et al. 2012). Additionally, exposure to domestic violence and substance abuse in minors can contribute to the development of antisocial behavior and delinquency (Perron 2013). These negative experiences can cause emotional trauma, lead to low self-esteem, and negatively affect the child's ability to form healthy relationships—all of which contribute to the likelihood of involvement in delinquent activities. Research has shown family conflict and dysfunction are risk factors for the emergence of juvenile delinquency (Perron 2013). This research echoes the articles described in this review, which tie exposure to trauma and adverse childhood experiences with increases in delinquent behavior and subsequent involvement with the juvenile justice system (Anderson and Walerych 2019; Garduno 2022). Children who experience physical or emotional abuse, neglect, parental intoxication or criminal behavior, or family violence are more prone to engage in delinquent behaviors (Henggeler et al. 2012; Perron 2013).

Additionally, children who live in high-conflict or abusive homes are more likely to have weaker family ties, less effective communication, and limited parental participation, all of which raise the likelihood that they would become delinquent (Young and Widom 2014). Furthermore, research has shown that parental criminal behavior, mental health issues, and substance addiction are linked to higher levels of family conflict and a higher risk of juvenile offending (Lantos et al. 2019; Leve et al. 2015). Familial strife not only has the ability to impact youth attachment to their family but also creates a cascade of negative emotions such as increasing risk for depression, difficulty recognizing positive emotions, antisocial behavior, and psychopathy (Young and Widom 2014). As a result, social and emotional issues like low self-esteem, impulsivity, and a lack of empathy may start to emerge leading to an increased likelihood of the emergence of juvenile crime (Young and Widom 2014). It is significant to remember that, although family conflict and dysfunction might influence criminality, they are not necessarily the root reasons, as some of this conflict and dysfunction is often associated with poverty, concentrated disadvantage, societal structures, and life chances (Dodson 2013).

**Neglect and Maltreatment.** Childhood maltreatment exposure is a major risk factor, and a reliable predictor, associated with youth involvement in the juvenile justice system (Leve et al. 2015), with about 3.5 million cases of suspected child abuse every year (Strathearn et al. 2020). Neglect and maltreatment can have significant impacts on juvenile delinquency through their disruption of the natural process of emotional development in the child (Young and Widom 2014). As a result, children of abusive parents tend to display more negative emotions than children of non-abusive parents and have deficits in emotional processing that extend into adulthood (Young and Widom 2014). Such deficits in positive

emotion recognition in adulthood can be attributed to a myriad of factors. However, two leading theories contend that these individuals have either developed negative worldviews or have experienced fewer positive emotions throughout their lifetimes, making it difficult for them to recognize positive emotions in adulthood (Young and Widom 2014). In comparison to children who grow up in stable and supportive circumstances, adolescents who experience neglect or maltreatment are more likely to engage in criminal conduct (Lantos et al. 2019). This finding is consistent with the articles reviewed in this systematic review, which found associations between neglect and adverse childhood experiences and juvenile delinquency and recidivism (Garduno 2022; Logan-Greene and Jones 2015; Logan-Greene et al. 2020; Ryan et al. 2013; Wilkinson et al. 2019). Neglect and maltreatment can cause emotional and psychological trauma in children, leading to the development of mental health problems such as depression, anxiety, and aggression (Strathearn et al. 2020). Beyond the psychosocial effects, youth raised in an abusive households tend to display lower academic performances and intelligence levels in childhood, deficits in cognitive development and educational and employment attainments, and sexual health problems (Strathearn et al. 2020; Young and Widom 2014). Youth removed from their households due to abuse or neglect also face negative outcomes, with youth placed in substitute care child welfare settings reporting higher arrest rates (Ryan 2012). Clearly, there is a need for improved responses to address abuse and neglect and foster healthier familial relationships.

**Individual and Family Mitigating Factors.** There are several factors that can help prevent youth from engaging in delinquent behavior and encourage positive development and healthy outcomes. As the articles in this section of this review note, access to family-based services, consistent discipline at home, strong familial relationships, and academic achievement are some protective factors against juvenile delinquency (Cavanagh and Cauffman 2017; Gearhart and Tucker 2020; Halgunseth et al. 2013; Ruch and Yoder 2018; Ryan 2012; Wilkinson et al. 2019). Children who are close to their parents and are actively involved in their families are less likely to act in a delinquent manner (Hoeve et al. 2012). Youth who are involved in school-based activities and who have a more engaged relationship with their schools are also less likely to exhibit antisocial behaviors (Henggeler et al. 2012). Participation in prosocial activities like athletics, music, or community work and constructive peer interactions have been shown to reduce the risk of juvenile delinquency (Henggeler et al. 2012). Individual qualities, including emotional stability, resiliency, and coping mechanisms, also contribute to an adolescent's propensity to engage in delinquent crime (Walker et al. 2019). Additionally, family-level factors can provide strong protection against juvenile delinquency if familial bonds are strong, or can have the opposite effect, as detailed in the section on family conflict and dysfunction (Halgunseth et al. 2013; Ruch and Yoder 2018; Wilkinson et al. 2019). Families that enforce consistent discipline and foster loving, high-quality relationships are associated with lower rates of juvenile delinquency (Cavanagh and Cauffman 2017; Halgunseth et al. 2013; Ruch and Yoder 2018; Wilkinson et al. 2019).

**Interventions.** Both family- and community-based interventions provide paths to target many of the risk factors for delinquency identified above and to foster more of the protective factors identified. Family-based interventions have demonstrated evidence-based success in lowering the risk of adolescent crime, including serious juvenile offenses, and recidivism (May et al. 2014). Additionally, family-based interventions can assist in strengthening the parent–child bond by building positive relationships through communication and problem solving, which can encourage prosocial conduct and lower the likelihood of recidivism (May et al. 2014). Strengthened family functioning and a reduction in affiliation with troubled peers are crucial steps in achieving positive outcomes for young offenders (Leve et al. 2015). Family and social support is an important buffer for youth involved in the court system, especially girls, as it fosters resilience in children and reduces delinquency (Anderson et al. 2021). These interventions also equip parents with the skills and awareness to enhance parental oversight and monitoring by providing them with support services, which may lessen the chance of juvenile delinquency (Leve et al. 2015).

While family-based interventions, especially family therapy, have been shown to be effective, there is a need to expand the focus of these interventions beyond just family therapy approaches to maximize their potential impact on juvenile delinquency. Existing community-based interventions leverage multi-sectoral engagement and utilize public spaces like parks and community centers to reach a greater number of individuals and to provide holistic services (Barrett and Janopaul-Naylor 2016; D'Agostino et al. 2020). More research is needed to better understand the most effective community-based interventions. By trying to address the root causes of the issue and helping the entire family or community build the skills to overcome it, these interventions can provide youth with the support and resources they need to overcome challenges.

Expansion of family-based interventions to the community level and development of strong community-based interventions are needed steps to increase access to effective treatments, as only 5% of serious juvenile offenders currently engage in such evidence-based treatments (Zajac et al. 2015). Given the multi-dimensional factors that affect delinquent behaviors among youth, interventions should consider the influence of family, peers, neighborhood, schools, and the larger community.

## 5. Limitations

The present study has a few limitations. The articles identified in this systematic review were gathered from two academic online databases: PubMed and Scopus. Therefore, our findings are limited to the peer-reviewed articles that appeared on these databases from 2012 to 2022. The use of only peer-reviewed articles may omit some grey literature, government reports, legal review papers, or additional commentary on the topic of juvenile delinquency and may introduce publication bias. Additionally, the search terms used focused primarily on family-based interventions. While some community-based interventions were identified within the resulting articles and were included in the review, the search was not comprehensive in terms of community-based interventions. Future reviews should examine the existing literature on community-based interventions more comprehensively. Lastly, variation across studies warrants caution in interpreting the findings of this review. For example, selected articles had sample sizes ranging from 15 to 19,833 participants and varied in the outcomes measured.

## 6. Conclusions

Articles examined in this systematic review explore the intricate web of factors that both promote and mitigate juvenile delinquency, as well as evaluating existing family- and community-based interventions aimed at reducing juvenile delinquency and recidivism. A significant amount of these risk and protective factors revolve around family dynamics with family conflict, dysfunction, neglect, and maltreatment emerging as predominant themes in juvenile delinquency literature. This review provides practical use of insights to inform policymaking, guide future research endeavors, and pinpoint actionable areas for improvement in correctional health settings for juveniles. Questions remain regarding which interventions are most effective and how such interventions can be scaled up to reach larger populations in need and prevent or reduce juvenile delinquency on a wider scale. This review provides a comprehensive overview of the factors and interventions that contribute to and reduce delinquency to help inform future research and identify potential targets for improvement.

**Author Contributions:** Conceptualization, H.Z.; methodology, H.Z.; software, R.V.; validation, H.Z.; investigation, A.A., R.V. and A.N.P.; resources, A.A., R.V. and A.N.P.; writing—original draft preparation, A.A., R.V. and A.N.P.; writing—review and editing, A.A., R.V., A.N.P. and H.Z.; supervision, H.Z.; project administration, A.A.; funding acquisition, H.Z. All authors have read and agreed to the published version of the manuscript.

**Funding:** This research was partially supported by the Bloomberg American Health Initiative (BAHI). Hossein Zare received an award from BAHI.

**Informed Consent Statement:** Not applicable.

**Data Availability Statement:** Not applicable.

**Conflicts of Interest:** The authors declare no conflict of interest.

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
