# Peer review of "Risk and Protective Factors and Interventions for Reducing Juvenile Delinquency: A Systematic Review"

_socsci, doi:10.3390/socsci12090474_

Round 1
Reviewer 1 Report
This paper, prepared by the Authors using the systematic review method, addresses risk factors and interventions to reduce juvenile delinquency.
Summary: A good introduction to the topic of the article. The Authors explain that juvenile delinquency is a huge problem in the United States, and the literature highlights the importance of early intervention and the role of the family in preventing juvenile delinquency. As a research method, the authors used the Preferred Reporting Items for Systematic Review and Meta-Analyses (PRISMA) structure for papers found in the PudMed and Scopus databases, including 29 peer-reviewed articles in English (for the period January 2012 - October 2022). The Authors analyzed the existing literature on risk factors, protective factors and interventions related to juvenile delinquency. They searched articles that discussed crime reduction and recidivism in the United States and coded them into four overarching themes: "family - conflict and dysfunction", "neglect and abuse", "individual and family mitigating factors", and "interventions at the family and community level". They found that family conflict, dysfunction, neglect and maltreatment are the two main causes of juvenile delinquency. Interventions to reduce crime should consider the influence of family, peers, neighborhood, school and local community.
Keywords: are correct and reflect the essence of the article well.
Introduction: The Authors explain from the literature that the United States has a higher youth incarceration rate than any other developed country. In contrast, youth incarcerated face higher rates of violence and fewer prospects for education and employment. He has poorer health records and is more likely to return to prison before age 30. Multi-system therapy is designed to enhance protective factors in crime prevention.
Material and method: The study was prepared using the systematic review method. The PRISMA framework outlines four steps for identifying articles to be used in a systematic review: identification, screening, eligibility and inclusion. The authors conducted a literature search using two electronic databases: Scopus and PubMed. Table 1 shows the search strategies used in each database. Inclusion criteria for peer-reviewed articles covering family dynamics, mitigating factors, and family and social interventions were detailed. The exclusion criteria for articles are also discussed in detail.
Results: The Authors developed the following themes: 1) family conflict and dysfunctions 2) neglect and abuse 3) individual and family mitigating factors 4) family and community interventions. Table 2 presents the study design, data sources, study populations, and key findings for each study.
The discussion discusses in detail the four main topics presented in this study based on the literature.
The Authors cited the limitations of this study, which result primarily from the choice of only two academic online databases: PubMed and Scopus, which may introduce publication bias.
Conclusions. The Authors emphasize that the article provides a comprehensive overview of the factors and interventions that contribute to crime reduction. Family and community-based alternatives have proven to be effective juvenile delinquency interventions. All literature items have been cited. The language of the work is understandable and correct in reception.
I rate this study well in terms of its content and innovation, and recommend it for publication.
Author Response
Reviewer 1
Dear Reviewer, we thank you so much for your positive feedback on our manuscript.
Comments and Suggestions for Authors
This paper, prepared by the Authors using the systematic review method, addresses risk factors and interventions to reduce juvenile delinquency.
Summary: A good introduction to the topic of the article. The Authors explain that juvenile delinquency is a huge problem in the United States, and the literature highlights the importance of early intervention and the role of the family in preventing juvenile delinquency. As a research method, the authors used the Preferred Reporting Items for Systematic Review and Meta-Analyses (PRISMA) structure for papers found in the PudMed and Scopus databases, including 29 peer-reviewed articles in English (for the period January 2012 - October 2022). The Authors analyzed the existing literature on risk factors, protective factors and interventions related to juvenile delinquency. They searched articles that discussed crime reduction and recidivism in the United States and coded them into four overarching themes: "family - conflict and dysfunction", "neglect and abuse", "individual and family mitigating factors", and "interventions at the family and community level". They found that family conflict, dysfunction, neglect and maltreatment are the two main causes of juvenile delinquency. Interventions to reduce crime should consider the influence of family, peers, neighborhood, school and local community.
Keywords: are correct and reflect the essence of the article well.
Dear Reviewer, we thank you so much for your positive feedback on our manuscript.
Author Response
Dear Reviewer:
We sincerely appreciate the afforded opportunity to revise our article in accordance with the guidance provided by the esteemed editors and reviewers. We wish to elucidate within this correspondence the manner in which we have addressed and incorporated their invaluable recommendations into the revised version of the article.
Sincerely,
Corresponding Author

Reviewer 3 Report
The paper has great relevance in terms of both theory and practice. Overall, the paper is well-written and based on relevant scientific resources. I have only some minor points which need to be clarified:
1) Abstract: The authors wrote: "We found that family conflict, dysfunction, neglect, and maltreatment are two primary reasons for juvenile delinquency." How do we mean "two" since they are four? In addition, the title is about interventions and I would suggest to focus on this in result and conclusion sections of abstract.
2) Intro: well-written, however, in the last paragraph, the aim of the study (i.e., the systematic review) should be more concretely and clearly given.
3) Results: Since the authors wanted to include both risk and protective factors (see title), I recommend that both should be mentioned in subheadings (now in most cases only risk factors are mentioned)
4) There are lots of overlapping parts in Results and Discussion sections. The authors should clairfy what is the aim of Discussion? More than interpreting the results?
5) Conclucisons: The readers may want to read more practical implications of the review.
In a word, the aim of the study as well as the conclusion should be more explicitely described to give a coherence to the otherwise well-written review.
Author Response

(The authors gave the same response as above.)

Round 2
Reviewer 2 Report
The manuscript has been improved.